# Evaluation of 3′-phosphate as a transient protecting group for controlled enzymatic synthesis of DNA and XNA oligonucleotides

Marie Flamme[1], Steven Hanlon[2], Irene Marzuoli[2], Kurt Püntener[2], Filippo Sladojevich[3] & Marcel Hollenstein [1✉]

Chemically modified oligonucleotides have advanced as important therapeutic tools as reflected by the recent advent of mRNA vaccines and the FDA-approval of various siRNA and antisense oligonucleotides. These sequences are typically accessed by solid-phase synthesis which despite numerous advantages is restricted to short sequences and displays a limited tolerance to functional groups. Controlled enzymatic synthesis is an emerging alternative synthetic methodology that circumvents the limitations of traditional solid-phase synthesis. So far, most approaches strived to improve controlled enzymatic synthesis of canonical DNA and no potential routes to access xenonucleic acids (XNAs) have been reported. In this context, we have investigated the possibility of using phosphate as a transient protecting group for controlled enzymatic synthesis of DNA and locked nucleic acid (LNA) oligonucleotides. Phosphate is ubiquitously employed in natural systems and we demonstrate that this group displays most characteristics required for controlled enzymatic synthesis. We have devised robust synthetic pathways leading to these challenging compounds and we have discovered a hitherto unknown phosphatase activity of various DNA polymerases. These findings open up directions for the design of protected DNA and XNA nucleoside triphosphates for controlled enzymatic synthesis of chemically modified nucleic acids.

[1] Institut Pasteur, Université de Paris Cité, CNRS UMR3523, Department of Structural Biology and Chemistry, Laboratory for Bioorganic Chemistry of Nucleic Acids, 28, rue du Docteur Roux, 75724 Paris Cedex 15, Paris, France. [2] Pharmaceutical Devision, Synthetic Molecules Technical Development, F. Hoffmann-La Roche Ltd, 4070 Basel, Switzerland. [3] Pharma Research and Early Development, Roche Innovation Center Basel, F. Hoffmann-La Roche Ltd, Grenzacherstrasse 124, 4070 Basel, Switzerland. ✉email: marcel.hollenstein@pasteur.fr

Xenonucleic acids (XNAs) are synthetic genetic polymers that differ from canonical nucleic acids mainly by the chemical composition of their sugar, phosphate, and nucleobase moieties[1–5]. The presence of chemical modifications on the scaffold of XNAs endows these biopolymers with enhanced properties compared to natural DNA and RNA. For instance, the presence of modified sugar units massively enhances their resistance against nuclease-mediated degradation which is an important prerequisite for the development of therapeutic oligonucleotides[6–10]. Similarly, the presence of additional functional groups on nucleobases or the installation of unnatural base pairs improve the binding and catalytic properties of nucleic acids[11–20]. So far, synthetic access to XNA oligonucleotides is granted by two different approaches: (i) automated solid-phase synthesis using phosphoramidite building blocks and (ii) polymerase-mediated synthesis with modified nucleoside triphosphates (dN*TPs). While the first approach permits to produce larger amounts of XNA oligonucleotides[3,21] it is limited in terms of size (less than 100 nucleotides) and functional group tolerance[6,22]. On the other hand, the chemoenzymatic method grants access to oligonucleotides of virtually any length[23] and permits in vitro selection experiments to identify XNA aptamers and XNAzymes[20,24–29]. However, this method also requires the use of specially engineered polymerases that are capable of copying DNA templates into XNA and then back into DNA[30–32] and all sites will contain the same type of modification. Recently, controlled enzymatic synthesis of DNA, a hybrid method combining elements of both approaches, has emerged and is raising increased attention. In this approach, nucleoside triphosphates are equipped with temporary 3'-protecting groups that can be removed after incorporation into a solid-phase bound primer sequence by a template dependent or independent DNA polymerase[33–38]. So far, most efforts focused on using the template-independent DNA polymerase Terminal deoxynucleotidyl Transferase (TdT) in conjunction with small, reversible protecting groups such as aminoalkoxyl[39] or 3'-O-azidomethylene[40] placed on the 3'-hydroxyl moiety of DNA nucleoside triphosphates (dNTPs). While this strategy culminated in the launch of a prototype of enzymatic DNA synthesizer by the biotechnology company DNA Script[41], controlled enzymatic synthesis is still mainly restricted to rather short oligonucleotides[42] and to deoxyribose chemistry exclusively. Herein, we have explored the possibility of expanding this method to the synthesis of XNA oligonucleotides. To do so, we have evaluated the use of 3'-phosphate as a simple, biocompatible protecting group for the controlled enzymatic synthesis of DNA and locked nucleic acid (LNA) oligonucleotides.

## Results

**Rationale and design**. The design of a reversible protecting group for controlled DNA and XNA synthesis involves a finely tuned balance between multiple factors. Indeed, polymerases have evolved as finely tuned enzymes capable of specifically recognizing canonical dNTPs or NTPs as substrates and to repel nucleotides with altered sugar moieties including those equipped with functional groups appended on the 2'/3'-OH groups[39,43]. Hence, the protecting group must be a rather small, preferably hydrophilic chemical entity that ensures substrate recognition by the polymerase and does not compromise its incorporation into DNA. In addition, the protecting group must be stable both upon storage of the nucleotide in buffered solution and during the polymerase-mediated catalytic step so as to prevent the simultaneous incorporation of multiple nucleotides. Concomitantly, cleavage of the protecting group should proceed in high yields under mild conditions so as not to damage the growing DNA/XNA chain and to permit synthesis of longer oligonucleotides. The installation of a 3'-phosphate group fulfills most of these criteria since it is not a

bulky, hydrophilic group that should be stable to hydrolysis under storage and synthesis and can easily be removed by the action of phosphatases. In order to evaluate the possibility of using a 3'-phosphate group to block the addition of nucleotides by polymerases we carried out primer extension (PEX) reactions with a 3'-phosphorylated primer with 10 different DNA polymerases and unmodified DNA dNTPs. Using a 31 nucleotide long template **T1** and a 15 nucleotide long, 5'-FAM-labeled primer **P1** equipped with a 3'-phosphate moiety[44] (see Supporting Information for sequence composition), all the reactions with the exception of those carried out with Therminator led to a negligible (i.e. <10% conversion) extension of the primer to full length or truncated products (Fig. 1). Importantly, treatment of primer **P1** with phosphatases such as the FastAP thermosensitive alkaline phosphatase allowed removal of the 3'-phosphate protecting group and facilitated polymerase-mediated DNA synthesis (Supplementary Fig. 1). Similar results were obtained with the TdT polymerase where the 3'-phosphorylated primer **P1** prevented the polymerase from adding dT nucleotides and treatment with FastAP thermosensitive alkaline phosphatase restored the tailing reaction capacity of the TdT (Supplementary Fig. 2). Lastly, we performed an Autodock simulation study using the reported X-ray structure of the ternary complex of mouse TdT with ssDNA and an incoming nucleotide (PDB 4I27). In each analysis, we replaced the incoming nucleotide with either 3'-phosphate LNA-TTP or 3'-phosphate-dTTP. This analysis revealed that both modified nucleotides were rather well tolerated within the active site of the TdT polymerase with favorable free energies (−16.94 kcal/mol and −17.40 kcal/mol for the protected dTTP and LNA-TTP, respectively; see Supplementary Figs. 3 and 4) comparable to that of unprotected LNA-TTP[44]. Taken together, these initial experiments suggest that the 3'-phosphate group can efficiently block DNA synthesis, can be removed by the action of phosphatases, and 3'-phosphorlyated dNTPs appear to be rather well tolerated within the active site of certain DNA polymerases at least according to docking experiments.

**Synthesis of 3'-phosphate-dTTP 5 and 3'-phosphate-LNA-TTP 10**. We next synthesized the 3'-phosphorylated versions of dTTP (3'-phos-dTTP **5**) and LNA-TTP (3'-phos-LNA-TTP **10**) to assess whether these modified nucleotides are compatible with controlled enzymatic DNA and XNA synthesis (Fig. 2 and Supplementary Figs. 46–69). To do so, we envisioned a common synthetic pathway that involved first conversion of the commercially available dT phosphoramidite **1** or the known LNA dT phosphoramidite **6**[45,46] to the corresponding H-phosphonates **2** and **7** using ETT as activator[47–49]. H-phosphonates **2** and **7** were then oxidized to the corresponding P(V) containing nucleotides with iodine under typical oxidation conditions used in solid-phase DNA synthesis. The DMTr masking groups of **3** and **8** were then removed under acidic conditions and the deprotected nucleoside analogues were converted to the expected dN*TPs **5** and **10** by application of the 4 step one pot method developed by Ludwig and Eckstein[50].

**Biochemical characterization of 3'-phosphorylated nucleotides 5 and 10**. With both 3'-phosphorylated nucleotides at hand, we set out to evaluate their substrate acceptance by DNA polymerases under PEX reaction conditions. To do so, we carried out PEX reactions using the **P1/T1** primer/template system along with 10 different DNA polymerases and with both 3'-phosphorylated nucleotide analogs (Fig. 3). When PEX reactions were conducted with Taq, the expected $n + 3$ product (corresponding to the addition of a dA, a dC, and one phosphorylated nucleotide) formed in moderate yields (~50%) in the presence of 3'-phos-dTTP **5** and 3'-phos-LNA-TTP **10**, which could be optimized to

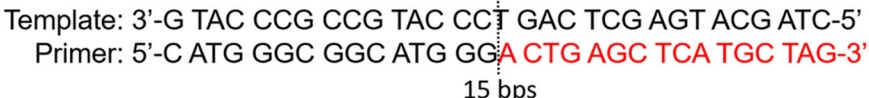

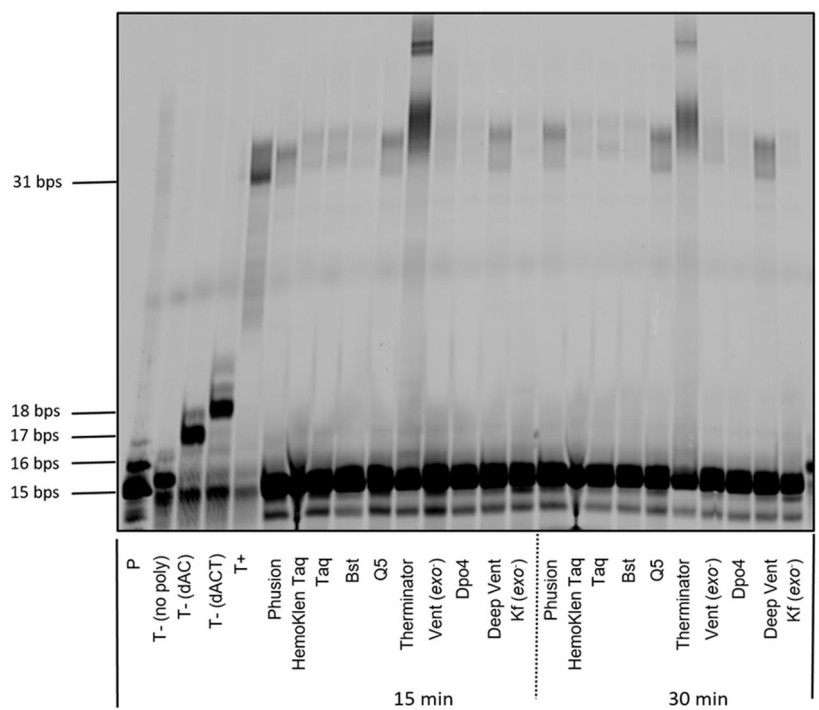

**Fig. 1 Blocking enzymatic synthesis with a 3′-phosphorylated primer.** Gel image (PAGE 20%) shows the result of the PEX reactions with 3′-phosphorylated primer P1 and template T1 with natural dNTPs and different DNA polymerases. Natural triphosphates were at a final concentration of 200 μM. The following quantities of polymerases and reaction conditions were used: Phusion (2 U), Hemo Klen Taq (8 reactions), Taq (5 U), *Bst* (8 U), Q5 (2 U), Therminator (2 U), Vent (*exo⁻*) (2 U): 60 °C, 15 min and 30 min; Dpo4 (2 U), Deep Vent (2 U): 55 °C, 15 min and 30 min; Kf (*exo⁻*) (5 U): 37 °C, 15 min and 30 min. Negative control (T−): No polymerase added to the mixtures or reactions only with dATP and dCTP or dATP, dCTP, and dTTP only. Positive control (T+): with all natural nucleotides and Taq polymerase. All reactions were incubated at adequate reaction temperatures for 1 h. **P** represents unreacted, 5′-FAM-labeled primer.

complete conversion of the primer to $n + 3$ product (Supplementary Fig. 5). Unexpectedly, all other polymerases extended the primer further and generated 22 nucleotide long oligonucleotides corresponding to $n + 7$ products. Reactions conducted with Therminator even led to the formation of full length products. Intrigued by these results, we analyzed the products stemming from the PEX reactions conducted with the combination of Taq polymerase and 3′-phos-dTTP **5** as well as that with Therminator and 3′-phos-LNA-TTP **10** by LCMS (Table 1, Supplementary Figs. 23–28, and Supplementary Note 1) using established protocols for products stemming from PEX reactions with natural and modified nucleotides[51,52]. This analysis clearly revealed that 1. no phosphorylated nucleotide was incorporated by polymerases and that 2. dA nucleotides were misincorporated opposite templating dAs instead of the modified triphosphates. Such a preference for dAMP misincorporation was further demonstrated when PEX reactions were performed in the presence of all natural dNTPs except for dTTP (Supplementary Fig. 6). Such a behavior was previously observed for the highly modified XNA nucleotide 7′,5′-bc-DNA since incorporation of the modified nucleotide by DNA polymerases proceeded with much lower efficiency than misincorporation of dAMP opposite templating dT residues[53]. Based on these considerations we next wondered if single incorporations might be observed when the 3′-phosphorylated analogs were used in the absence of competitors such as dATP. Hence, we carried out PEX reactions with template **T2** that contains a stretch of dA nucleotides immediately 3′-downstream

of the corresponding primer **P1** as well as with template **T3** which was designed as a universal template for controlled DNA synthesis[34].

The PEX reactions carried out with dN*TPs **5** and **10** individually and template **T2** are shown in Fig. 4. Analysis of the reaction products by gel electrophoresis revealed that both nucleotides were seemingly well accepted by a number of DNA polymerases since bands corresponding to $n + 1$ and $n + 2$ and sometimes even $n + 3$ and $n + 4$ products could be observed with both nucleotides with the polymerases HemoKlenTaq, Bst, Therminator, Vent (*exo⁻*), and Kf (*exo⁻*). Similar results were obtained when the universal template **T3** was used instead of **T2** (Supplementary Fig. 7). Here as well, we performed an LCMS analysis of the PEX reaction products obtained with the **P1/T3** system in order to try to understand the origin of these multiple incorporation events. The results obtained with the $n + 1$ and $n + 2$ products with both modified nucleotides are summarized in Table 2 (also see Supplementary Figs. 29–40).

This analysis clearly reveals that both modified nucleotides are successfully incorporated when no competitor such as dATP is present. On the other hand, all observed products correspond to the addition of one or two nucleotides onto primer **P1** but *without* the presence of the 3′-phosphate protecting group. These results suggest that both A-family (e.g. Taq) and B-family (e.g. Vent (*exo⁻*)) DNA polymerases are capable of removing the 3′-phosphate protecting group either at the level of the incoming nucleotide or once installed on the extended primer.

**Fig. 2 Synthesis of 3′-phosphate-dTTP 5 and 3′-phosphate-LNA-TTP 10. A** Synthetic scheme for the synthesis of nucleotide **5** and (**B**) synthetic pathway leading to nucleotide **10**. Reagents and conditions: (i) ETT, CH₃CN, H₂O (9:1), rt, 10 min, quantitative for both **2** and **7**; (ii) I₂, pyridine, H₂O (9:1), rt, 60 min, quantitative for both **3** and **8**; iii) TFA, DCM, rt, 60 min, **4** (84%), **9** (97%); iv) 2-chloro-1,3,2-benzodioxaphosphorin-4-one, pyridine, dioxane, rt, 45 min; 2. (nBu₃NH)₂ H₂P₂O₇, DMF, nBu₃N, rt, 45 min; 3. I₂, pyridine, H₂O, rt, 30 min; 4. NH₄OH, MeNH₂ (1:1), rt, 2 h, 12% over 4 steps for both **5** and **10**.

We next questioned whether the 3′-phosphate protected nucleotides are accepted as substrates by other polymerase families and whether the phosphate protecting group is also removed by these polymerases. To do so, we performed template-independent PEX reactions using the X-family DNA polymerase TdT along with the 3′-protected nucleotides **5** and **10**. In addition, we supplemented the reaction mixtures with three different M²⁺ cofactors since the metal preference of TdT is not very strict[33]. After significant optimization of the reaction conditions with 3′-phos-dTTP **5**, ~50% conversion of 5′-FAM-labeled primer **P2** (Supporting Information for sequence composition)[54] into the corresponding $n + 1$ product was observed in the presence of Mn²⁺ alone or together with Mg²⁺ (Fig. 5). On the other hand, 3′-phos-LNA-TTP **10** was not well accepted as a substrate by the TdT polymerase since very modest yields (10–20%) of $n + 1$ product formed even after long reaction times or when the feed ratio of monomers (i.e., modified triphosphates) to initiator (i.e., primer) was increased (Supplementary Fig. 8)[55].

Collectively these results demonstrate that 3′-phosphate protected nucleotides are not very good substrates for DNA polymerases, particularly for family X polymerases such as the TdT. In the absence of competing nucleotides such as dATP, these nucleotides are readily incorporated into DNA by various family A and B polymerases but at the expense of an incomplete blocking activity of the 3′-phosphate group presumably due to the inherent esterase/phosphatase activity of several DNA polymerases. Certain DNA polymerases were recently shown to display an esterase activity (see Discussion) and hence a phosphatase activity is not totally unexpected. Even though commercially available DNA polymerases are certified by the supplier to display less than 0.0001 unit of alkaline phosphatase activity (New England Biolabs), we have performed an MS analysis on Kf (exo⁻) which confirmed the

absence of any contaminants including phosphatases (Supplementary Figs. 89 and 90, Supplementary Table 1, Supplementary Note 2, and Supplementary Discussion).

**Effect of charge: 3′-cyanoethyl-phosphate protecting group**. We first hypothesized that the poor acceptance of nucleotides **5** and **10** as polymerase substrates might be ascribed to the presence of the negative charges on the 3′-phosphate moiety. Hence, we rationalized that an additional phosphoester bond on the 3′-protecting group might reduce this negative charge and improve the substrate tolerance. Such an additional ester linkage is readily available if the β-cyanoethyl protecting group of the original phosphoramidites is not removed by cleavage with ammonia. Moreover, docking experiments revealed that dTTP and LNA-TTP equipped with 3′-β-cyanophosphate groups fitted well into the active site of the TdT polymerase (Supplementary Figs. 9 and 10). The free energy for the docking of 3′-β-cyanophosphate-dTTP into the active site of the TdT was comparable to that of the corresponding 3-phosphate nucleotide (−16.65 kcal/mol) while a much more favorable free energy was obtained with 3′-β-cyanophosphate-LNA-TTP (−18.18 kcal/mol). We thus converted analogues **4** and **9** into the corresponding triphosphates **11** and **12** by the application of the Ludwig Eckstein protocol (Fig. 6 and Supplementary Figs. 70–76).

With nucleotides **11** and **12** at hand, we evaluated their substrate capacity for DNA polymerases under PEX reactions with templates **T1** and **T2** as well as the possibility of using these analogs for TdT-mediated extension reactions. Gel analysis of all the products obtained from PEX reactions revealed that both nucleotides acted as poor substrates for DNA polymerases (Supplementary Figs. 11–14). Indeed, we either observed only

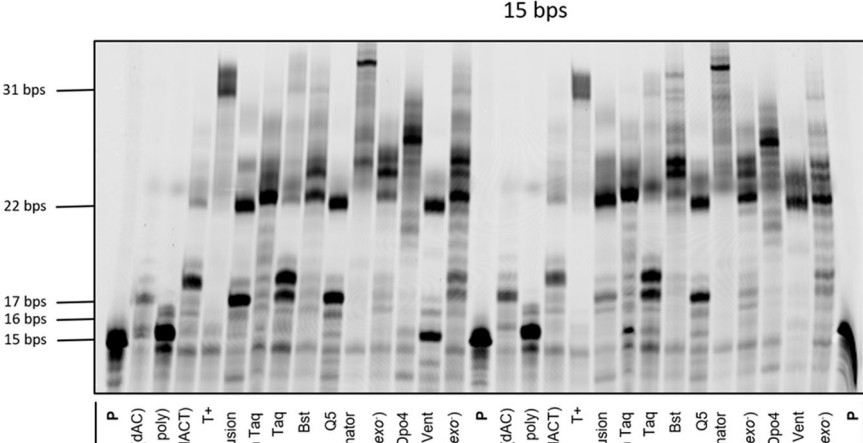

**Fig. 3 Evaluation of modified nucleotides 5 and 10 under PEX reaction conditions.** Gel image (PAGE 20%) shows the results of PEX reactions with 3′-phos-dTTP **5** and 3′-phos-LNA-TTP **10** with 5′-FAM-labeled primer **P1** (devoid of a 3′-phosphate moiety), template **T1**, and various DNA polymerases. Natural and modified triphosphates were at a final concentration of 200 μM. The following quantities of polymerases were used: Phusion (2 U), Hemo Klen Taq (8 reactions), Taq (5 U), *Bst* (8 U), Q5 (2 U), Therminator (2 U), Vent (*exo⁻*) (2 U): 60 °C, 1 h; Dpo4 (2 U), Deep Vent (2 U): 55 °C, 1 h; Kf (*exo⁻*) (5 U): 37 °C, 1 h. Negative control (T−): No polymerase added to the mixtures or reactions with only dATP and dCTP or dATP, dCTP, and dTTP only. Positive control (T+): with all natural nucleotides and Taq polymerase. All reactions were incubated at adequate reaction temperatures for 1 h. **P** represents unreacted, 5′-FAM-labeled primer.

**Table 1 Summary of LCMS analysis of PEX reaction products.**

| PEX reaction condition | $m/z$ found | $m/z$ calculated | Nature of product |
|---|---|---|---|
| Taq and 3′-phos-dTTP **5** | 6124.092 | 6124.091 | **P1** + 2dA+dC |
| Therminator and 3′-phos-LNA-TTP **10** | 10185.786 | 10185.788 | Fully extended primer **P1** with dA incorporated instead of dT |

very little $n + 1$ formation or multiple incorporation events were detected at longer reaction times, presumably due to the loss of the β-cyanoethyl protecting group caused by β-elimination in the lower pH of the polymerase buffers. On the other hand, highly contrasting results were obtained when both nucleotides were assayed with the TdT polymerase. Indeed, while the TdT polymerase did not accept the blocked canonical nucleotide **11** (Supplementary Fig. 15), large product distributions were observed when LNA nucleotide **12** was used as substrate (Fig. 7). These results are surprising because i) LNAs are poor substrates for the TdT and usually terminate synthesis after the addition of a single nucleotide[44,56]; (ii) nucleotide **11** is not recognized as a substrate by the TdT and the primer is not extended by the polymerase; iii) the protecting group is removed either by the polymerase or in the reaction medium.

In order to shed some light into these results, we first performed TdT reactions with nucleotide **11** followed by the addition of 3′-unblocked LNA-TTP (Supplementary Fig. 16). This experiment clearly revealed that only a single LNA nucleotide was incorporated by the polymerase, suggesting that nucleotide **11** was not recognized by the enzyme. Next, we analyzed the products stemming from the TdT-mediated tailing reaction in conjunction with LNA nucleotide **12** by LCMS (Table 3 and Supplementary Figs. 43–S45). This analysis reveals that the intermediate bands as well as the $n_i$ and $n_{i+1}$ products

correspond to different chemical entities. In particular, bands corresponding to single or multiple addition events consist of the primer with one or multiple dehydrated LNA nucleotides devoid of any protecting groups. Such dehydration events have been observed in MS analysis of modified nucleotides[57]. In addition, bands that run between these bands correspond to similar species albeit with an additional $\Delta m/z$ of 15 compared to the parent bands. Such a $\Delta m/z$ is typically observed with misincorporation events (e.g. incorporation of dG instead of dA opposite templating dT or dC instead of dT opposite dA) under standard PEX reaction conditions[58]. However, under our experimental conditions, only modified triphosphate **12** was present as substrate. Moreover, similar double-banding events such as that displayed in Fig. 7 have already been described in the past for TdT primer extension reactions carried out in conjunction with sugar and 5′-phosphate-modified nucleotides[59]. This gel pattern was ascribed to the capacity of TdT to phosphorylate (and phosphonylate) oligonucleotides. While the double-banding pattern appears similar, the LCMS analysis of products does not fit with such a phosphorylation event. Such a $\Delta m/z$ might potentially be connected to deprotection of the β-cyanoethyl moiety and the concomitant addition of the resulting acrylonitrile on a nucleobase. Hence, in order to shed more light into the nature of these products we first analyzed the stability of the cyanoethyl group on nucleotide **11** in a TdT-mediated reaction

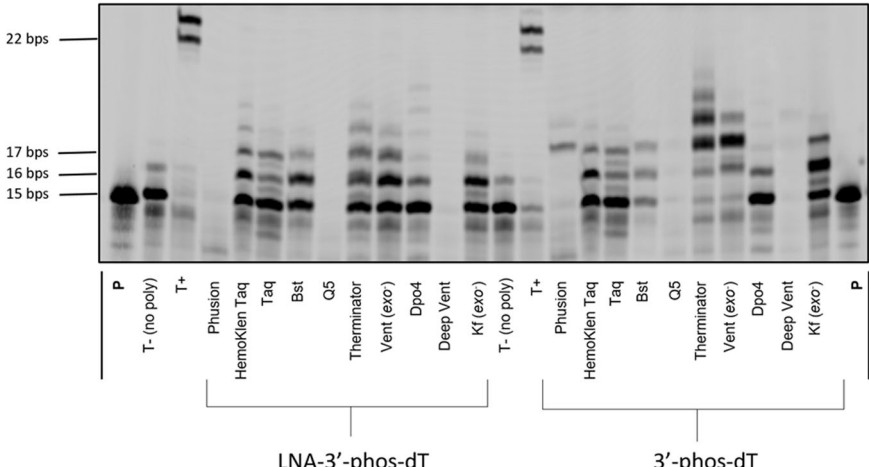

Template: 3'-G TAC CCG CCG TAC CCA AAA AAA-5'
Primer: 5'-C ATG GGC GGC ATG GGT TTT TTT-3'

**Fig. 4 Primer extension reactions with modified nucleotides 5 and 10 on a template containing a terminal stretch of dA nucleotides.** Gel image (PAGE 20%) shows the results of PEX reactions with 3'-phos-dTTP **5** and 3'-phos-LNA-TTP **10** with the **P1/T2** primer/template system and various DNA polymerases. Modified triphosphates were at a final concentration of 200 µM. The following quantities of polymerases and reaction conditions were used: Phusion (2 U), Hemo Klen Taq (8 reactions), Taq (5 U), *Bst* (8 U), Q5 (2 U), Therminator (2 U), Vent (*exo⁻*) (2 U): 60 °C, 1 h; Dpo4 (2 U), Deep Vent (2 U): 55 °C, 1 h; Kf (*exo⁻*) (5 U): 37 °C, 1 h. Negative control (T−): No polymerase added. Positive control (T+): with all natural nucleotides and Taq polymerase. All reactions were incubated at adequate reaction temperatures for 1 h. **P** represents unreacted, 5'-FAM-labeled primer.

**Table 2 Summary of the results from the LCMS analysis of the PEX reaction products obtained on the P1/T3 system and with modified nucleotides.**

| Reaction conditions | *m/z* calculated | *m/z* found |
|---|---|---|
| 3'-phos-dTTP 5 and Vent (*exo⁻*); $n+1$ product | 5512.9757 | 5512.978 |
| 3'-phos-dTTP 5 and Kf (*exo⁻*); $n+1$ product | 5512.9757 | 5512.979 |
| 3'-phos-LNA-TTP 10 and Vent (*exo⁻*); $n+1$ product | 5522.9600 | 5522.963 |
| 3'-phos-LNA-TTP 10 and Vent (*exo⁻*); $n+2$ product | 5836.9904 | 5836.994 |
| 3'-phos-LNA-TTP 10 and Kf (*exo⁻*); $n+1$ product | 5522.9600 | 5522.963 |

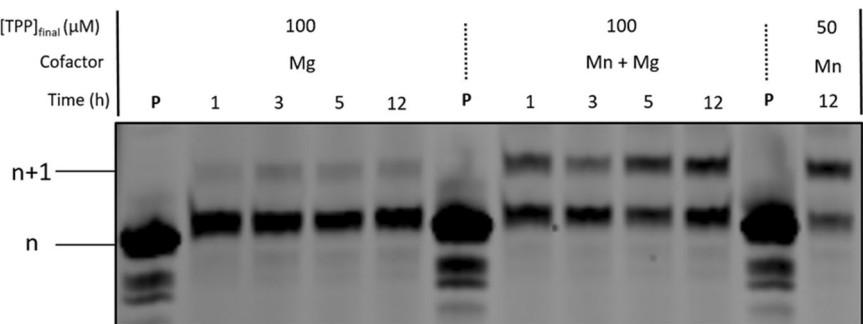

**Fig. 5 Evaluation of modified nucleotide 5 under TdT-mediated reactions.** Gel image (PAGE 20%) shows the results of the TdT-mediated extension reactions with 3'-phos-dTTP **5** and 5'-FAM-labeled primer **P2** (20 pmoles). Reaction mixtures contained TdT (10 U), triphosphate at 100 or 50 µM concentration, magnesium and/or manganese cofactors at 1 mM concentration, and were incubated at 37 °C for given reaction times. **P** represents unreacted, 5'-FAM-labeled primer.

using LCMS (Supplementary Figs. 17–19). This analysis revealed that even after 12 h of incubation, the chemical integrity of nucleotide **11** was not altered and no loss of the cyanoethyl group could be detected under these conditions. On the other hand, when LNA-TTP[44] was incubated with acrylonitrile prior to the extension reaction, only the expected $n+1$ products could be observed suggesting that the emergence of additional

products might arise via a different, yet unidentified mechanism (Supplementary Fig. 20).

**Enhancing resistance against hydrolysis: 3'-thiophosphate group**. The addition of an additional cyanoethyl moiety reduced the charge present on the 3'-phosphate blocking groups but

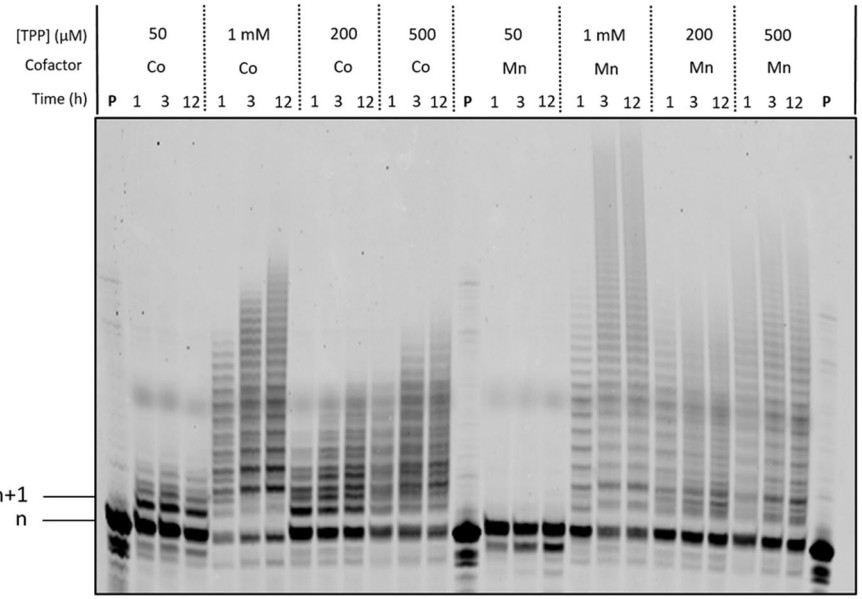

**Fig. 6 Synthesis of 3′-β-cyanophosphate-dTTP 11 and 3′-β-cyanophosphate-LNA-TTP 12. A** Synthetic scheme for the synthesis of nucleotide **11** and (**B**) synthetic scheme leading to nucleotide **12**. Reagents and conditions: (i) 2-chloro-1,3,2-benzodioxaphosphorin-4-one, pyridine, dioxane, rt, 45 min; 2. ($n$Bu$_3$NH)$_2$ H$_2$P$_2$O$_7$, DMF, $n$Bu$_3$N, rt, 45 min; 3. I$_2$, pyridine, H$_2$O, rt, 30 min, 13% over 3 steps for both **11** and **12**.

**Fig. 7 Evaluation of modified nucleotide 12 under TdT-mediated reactions.** Gel image (PAGE 20%) shows the results of the TdT-mediated extension reactions with 3′-β-cyanophosphate-LNA-TTP **12** and primer **P2** (20 pmoles). Reaction mixtures contained TdT (10 U), triphosphate at various, given concentrations, Co$^{2+}$ (0.25 mM) or Mn$^{2+}$ (1 mM) cofactors, and were incubated at 37 °C for given reaction times. **P** represents unreacted, 5′-FAM-labeled primer.

**Table 3 LCMS analysis of reaction products obtained with the TdT, 5′-FAM-labeled primer P2, Co$^{2+}$, and 3 h of reaction at 37 °C.**

| Product | *m/z* found (Da) | Difference (Da)[a] | Interpretation |
|---|---|---|---|
| Intermediate 1[b] | 6571.1710 | 329.0530 | Primer + 1LNA- 1H$_2$O + 15 Da |
| $n+1$ | 6556.1480 | 314.0300 | Primer + 1LNA- 1H$_2$O |
| Intermediate 2 | 6885.2000 | 643.0820 | Primer + 2LNA- 2H$_2$O + 15 Da |
| $n+2$ | 6870.1770 | 628.0590 | Primer + 2LNA- 2H$_2$O |
| Intermediate 3 | 7199.2290 | 957.1110 | Primer + 3LNA- 3H$_2$O + 15 Da |
| $n+3$ | 7184.2070 | 942.0890 | Primer + 3LNA- 3H$_2$O |
| Intermediate 4 | 7513.2590 | 1271.1410 | Primer + 4LNA- 4H$_2$O + 15 Da |
| $n+4$ | 7498.2390 | 1256.1210 | Primer + 4LNA- 4H$_2$O |
| Intermediate 5 | 7827.2870 | 1585.1690 | Primer + 5LNA- 5H$_2$O + 15 Da |
| $n+5$ | 7812.2670 | 1570.1490 | +5LNA- 5H$_2$O |

[a]Δ*m/z* with the unmodified primer **P2** (6242.1180).
[b]bands located between those of the $n_i$ and $n_{i+1}$ products.

introduced a steric bulk that precludes efficient incorporation of the resulting nucleotides by most polymerases. Hence, we questioned whether a minimal perturbation of the phosphate moiety such as the substitution of an oxygen moiety by a sulfur atom could improve the substrate capacity of 3'-phosphate-

modified LNA nucleotides. We rationalized that the introduction of a sulfur atom could decrease the capacity of polymerases at hydrolyzing the phosphate moiety since reaction at P = S centers is slower than for the native P = O centers[60,61] and concentrating the negative charge on sulfur could increase

**Fig. 8 Synthesis of 3′-thiophosphate-LNA-TTP 15 and 3′-β-cyano-thiophosphate-LNA-TTP 16.** Reagents and conditions: (i) Beaucage reagent, pyridine, rt, 10 min, quantitative; (ii) TFA, DCM, rt, 60 min, quantitative; (iii) 2-chloro-1,3,2-benzodioxaphosphorin-4-one, pyridine, dioxane, rt, 45 min; 2. ($n$Bu$_3$NH)$_2$ H$_2$P$_2$O$_7$, DMF, $n$Bu$_3$N, rt, 45 min; 3. I$_2$, pyridine, H$_2$O, rt, 30 min; 4. NH$_4$OH, MeNH$_2$ (1:1), rt, 2 h, 5% over 4 steps; iv) 2-chloro-1,3,2-benzodioxaphosphorin-4-one, pyridine, dioxane, rt, 45 min; 2. ($n$Bu$_3$NH)$_2$ H$_2$P$_2$O$_7$, DMF, $n$Bu$_3$N, rt, 45 min; 3. I$_2$, pyridine, H$_2$O, rt, 30 min, 7% over 3 steps.

interactions with polymerases[62]. Docking experiments comforted these assumptions since the sulfur atom is predicted to interact mainly with an arginine of the active site of the TdT and the overall free energy is very favorable (−17.50 kcal/mol; see Supplementary Fig. 21).

Synthesis of the 3′-thiophosphate-bearing nucleotide **15** is highlighted in Fig. 8 and makes use of our recently developed method for the synthesis of thiophosphates with the Beaucage reagent[44]. Briefly, *H*-phosphonate **7** (Fig. 2) is oxidized to the corresponding P(V) nucleotide **13** using the Beaucage reagent. After deprotection of the DMTr group, the 3′-phosphorothioate nucleotide **14** is converted to the corresponding 5′-triphosphate using the Ludwig-Eckstein approach (Supplementary Figs. 77–88).

The substrate acceptance of nucleotide **15** for DNA polymerases was investigated in PEX reactions and TdT-mediated tailing reactions (Fig. 9). Clearly, the presence of a 3′-thiophosphate moiety does not improve the substrate acceptance by polymerases since a similar product distribution as with LNA nucleotide **10** (Fig. 3) is observed during PEX reactions with primer **P1** and template **T1** (Fig. 9A). With the TdT, formation of the expected $n + 1$ product resulted but in low yields (~20%) and longer reaction times led to the appearance of additional bands, presumably stemming from hydrolytic degradation of the primer (Fig. 9B). These results suggest that the presence of a sulfur atom on the 3′-phosphate moiety does not improve the substrate acceptance by polymerases since misincorporation events might be favored even though the predicted hydrolysis of the protecting group might be reduced. Next, we synthesized nucleotide **16** (Fig. 8) which presents both a sulfur and a β-cyanoethyl moiety on the terminal 3′-phosphate group in order to evaluate whether the combination of a sulfur atom and a reduction of the negative charge could improve the incorporation efficiency. However, similar PEX reactions conducted with primer **P1** and template **T1** (Supplementary Fig. 22) and with the TdT (Supplementary Fig. 23) did only show marginal improvements compared to the incorporation efficiency of the parent compound **15**. Interestingly, the presence of the β-cyanoethyl moiety did not lead to multiple incorporation events when the TdT was used as polymerase as was the case for nucleotide **12** that has a P = O center rather than a P = S. Docking experiments reflect these results since a lower free energy (−16.81 kcal/mol) was calculated and unfavorable positioning of the 3′-protecting group within the active site of the polymerase were detected (Supplementary Fig. 24).

## Discussion

Controlled enzymatic synthesis of DNA, RNA, and XNAs represents an interesting and versatile alternative to chemical, phosphoramidite-based synthesis since in principle it is devoid of sequence length limitations and should be more tolerant to chemical modifications on nucleotides and oligonucleotides. This approach would be highly beneficial in a number of practical applications including storage of digital information[63–66], assembly of synthetic genes and genomes[67,68], or functional RNA oligonucleotides[69]. However, despite recent progress and increased interest in this methodology, no universal blocking group has been identified yet that allows synthesis of longer stretches of nucleic acids, particularly of XNAs. This difficulty resides in a delicate balance between steric bulk, robustness, and lability of a protecting group which is required to ensure substrate recognition of nucleotides by polymerases, efficient incorporation into oligonucleotides, and high yielding coupling and deprotection steps. In this context, we have explored the possibility of using 3′-phosphate as a temporary protecting group. Indeed, phosphate is ubiquitously used in nature for transient protection/modifications of proteins but also of nucleotides and oligonucleotides. Phosphate moieties are robust but can easily be removed by phosphatases and do not introduce a massive steric bulk into scaffolds. Nucleotides bearing 3′-phosphate moieties, however, are poor substrates for A- and B-family DNA polymerases since misincorporation of dAMP moieties is favored to incorporation of such modified nucleotides. This poor substrate acceptance by polymerases might be ascribed to the presence of two negative charges—even though partially masked by interaction with mono- or divalent metal cations or by interactions with residues of side chains of the active sites of polymerases. A similar accumulation of negative charge at the 3′-end of nucleotides might also explain the inhibitory effect of magic spot nucleotides or alarmones (i.e. guanosine-3′,5′-bis(diphosphate) ppGpp and guanosine-3′-diphosphate-5′-triphosphate pppGpp) even though these compounds have never been assayed in conjunction with DNA polymerases[70,71].

When reaction mixtures were supplemented with 3′-phosphate containing nucleotides alone, multiple incorporation events were observed which results from abstraction of the protecting group. Since 3′-phosphorylated primers cannot be extended by polymerases, we ascribe these multiple incorporation events to a moonlighting, phosphatase activity of polymerases *directly* at the level of the incoming, modified nucleotides. This observation is

**A)**

Template: 3'-G TAC CCG CCG TAC CCT GAC TCG AGT ACG ATC-5'
Primer: 5'-C ATG GGC GGC ATG GGA CTG AGC TCA TGC TAG-3'

15 bps

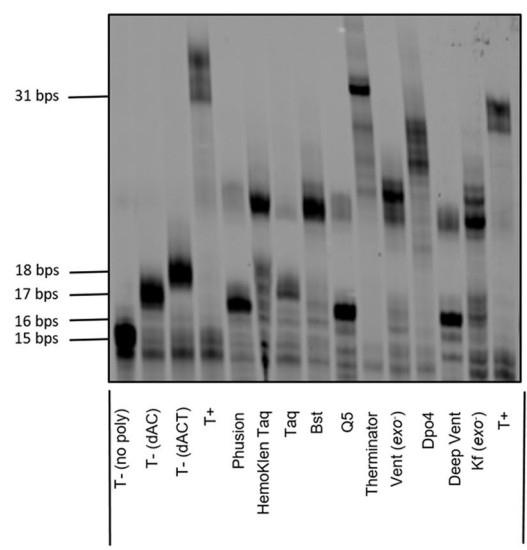

**B)**

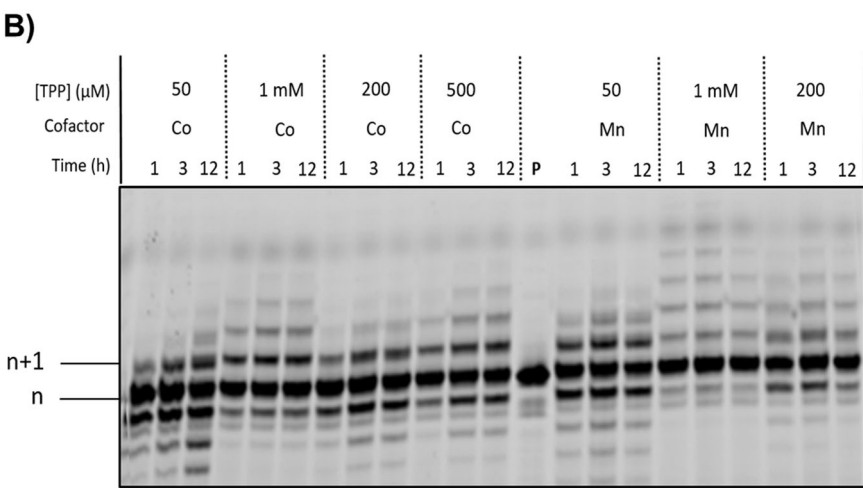

**Fig. 9 Biochemical evaluation of nucleotide 15.** Gel image (PAGE 20%) shows the analysis of the products stemming from (**A**) PEX reactions with various DNA polymerases and the **P1/T1** system and (**B**) TdT-mediated extension reactions with 5'-FAM-labeled primer **P2**. PEX reaction contained natural and modified triphosphates at 200 μM. The following quantities of polymerases were used: Phusion (2 U), Hemo Klen Taq (8 reactions), Taq (5 U), Bst (8 U), Q5 (2 U), Therminator (2 U), Vent (exo−) (2 U): 60 °C, 30 min; Dpo4 (2 U), Deep Vent (2 U): 55 °C, 30 min; Kf (exo−) (5 U): 37 °C, 30 min. Negative control (T−): No polymerase added to the mixtures or reactions with only dATP and dCTP or dATP, dCTP, and dTTP only. Positive control (T+): with all natural nucleotides and Taq polymerase. All reactions were incubated at adequate reaction temperatures for 1 h. TdT reaction mixtures contained TdT (10 U), triphosphate at various, given concentrations, Co$^{2+}$ (0.25 mM) or Mn$^{2+}$ (1 mM) cofactors, and were incubated at 37 °C for given reaction times. **P** represents unreacted, 5'-FAM-labeled primer.

not totally unexpected since various polymerases including HIV-RT, Sequenase[72], an exonuclease-deficient variant of the archaeal B-family 9°N DNA polymerase[73], and the large fragment of the A-family DNA polymerase from *Bacillus stearothermophilus* (BF)[74] possess an efficient 3'-esterase activity once the ester group is installed on the extended primer. Recently, DNA polymerase I fragment (Klenow) was shown to possess a phosphatase activity at the level of nucleotides but this activity consisted in the removal of one or two phosphate groups from 5'-triphosphate entities and only in the strict presence of RNA[75,76]. Lastly, some DNA polymerases (mainly belonging to family X polymerases) such as involved in repair pathways recruit Polymerase Histidinol Phosphatase (PHP) domains to mediate phosphatase activity[77]. While additional work will be necessary to pinpoint the site involved in such activity and to unravel its mechanism, this phosphatase activity of polymerases is unprecedented and further underscores the capacity of polymerases to act as enzymes with promiscuous activities.

## Conclusions

Controlled enzymatic synthesis represents an alluring alternative to traditional synthetic methods for the generation of wild type and modified oligonucleotides. While intense research has been dedicated to the development of methods and protecting groups suitable for DNA synthesis, little or no efforts have been devoted to similar strategies but for RNA or XNAs. In this context, we have explored the possibility of using phosphate as a transient 3′-blocking group for controlled enzymatic synthesis of DNA and LNA containing oligonucleotides. While this protecting group does not appear suitable for our approach despite meeting most of the required criteria, an unexpected and unprecedented moonlighting activity of various family A and B DNA polymerases was discovered. These results will allow us to refine the design and the chemical nature of other 3′-protecting groups to be explored for the controlled synthesis of XNA oligonucleotides and might have repercussions in understanding the mechanism of alarmones and the effect of phosphorylation of nucleotides in complex systems.

## Methods

**General protocol of TdT-mediated tailing reactions.** Primer **P2** (20 pmol) is incubated with the modified nucleoside triphosphate (200 μM) with a suitable metal cofactor (0.25 mM $Co^{2+}$, 1 mM $Mn^{2+}$, or 1 mM $Mg^{2+}$) and the TdT polymerase (10 U) in 1X reaction buffer (supplied with the polymerase; 10 μL final volume) at 37 °C for given reaction times. The reaction mixtures were then purified by Nucleospin columns and quenched by the addition of an equal volume of loading buffer (formamide (70%), ethylenediaminetetraacetic acid (EDTA, 50 mm), bromophenol (0.1%), xylene cyanol (0.1%)). The reaction products were then resolved by electrophoresis (PAGE 20%) and visualized by phosphorimager analysis.

**General procedure for primer extension reactions.** The template (15 pmol) was annealed to its complementary primer (10 pmol) by heating to 95 °C and slowly (over 30 min) cooling down to room temperature. The annealed oligonucleotides were then supplemented with modified and/or natural dNTPs (all 200 μM final concentrations) and polymerase (2 U) in 1X reaction buffer. The reaction mixtures were then incubated at the recommended temperature for given amounts of time. The reaction mixtures were then purified by Nucleospin columns and quenched by the addition of an equal volume of loading buffer (formamide (70%), ethylenediaminetetraacetic acid (EDTA, 50 mm), bromophenol (0.1%), xylene cyanol (0.1%)). The reaction products were then resolved by electrophoresis (PAGE 20%) and visualized by phosphorimager analysis.

**Chemical syntheses.** Detailed protocols for the synthesis of all nucleoside and nucleotide analogs can be found in the Supporting Information of this article.

**Docking experiments.** AutoDock version 4.2 was used for the docking simulation[78]. The TdT enzyme file was prepared using published coordinates (PDB 4I27). The magnesium atom was retained within the protein structure. A charge of +2 and a solvation value of −30 were manually assigned to the Mg atom. The molecules files were built on Biovia Discovery Studio® 4.5 and saved as pdb files. The docking area was assigned visually around the presumed active site. A grid of 40 Å x 40 Å x 40 Å with 0.497 Å spacing was calculated around the docking area using AutoGrid. We selected the Lamarckian genetic algorithm (LGA) for ligand conformational searching, which evaluates a population of possible docking solutions and propagates the most successful individual solution from each generation into the subsequent generation of possible solutions.

For each compound, the docking parameters were as follows: trial of 20 dockings, population size of 150, random starting position and conformation, translation step ranges of 1.5 Å, rotation step ranges of 35°, elitism of 1, mutation rate of 0.02, crossover rate of 0.8, local search rate of 0.06 and 2,500,000 energy evaluations. The docking method was first evaluated by redocking the corresponding ligand of the PDB structure and then docking of the molecules of interest in the TdT active site. The conformation of the obtained results was inspected and compared to the literature and crystal structures. The docking results from each of the compounds were clustered on the basis of the root-mean-square deviation (rmsd) of the Cartesian coordinates of the atoms and were ranked on the basis of free energy of binding. The top-ranked compounds were visually inspected for correct chemical geometry.

**Reporting summary.** Further information on research design is available in the Nature Research Reporting Summary linked to this article.

## Data availability

The authors declare that all data supporting the findings of this study are available within the article and the Supplementary Information.

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

## Acknowledgements

The authors thank Institut Pasteur (starting funds to M.H.) and Roche for financial support. We would like to thank Siegfried Stolz and Andreas Stämpfli (Roche) for help with the LCMS analysis and Martin Olbrich (Roche) for help with the manuscript preparation and for fruitful discussions. We would like to thank Chiara Figazzolo (Institut Pasteur) for running additional HPLC experiments, Olena Mayboroda and Sébastien Brier (Institut Pasteur) for the MS analysis of the Klenow polymerase.

## Author contributions

M.F. performed all the chemical synthesis, chemical and biochemical characterization, docking experiments, and contributed to the writing of the manuscript. I.M. participated in and performed docking experiments. S.H., K.P., F.S., and M.H. performed the study design and conception and analyzed data. S.H. and K.P. contributed to the writing of the manuscript. M.H. wrote the manuscript and critically revised the manuscript.

## Competing interests

The authors declare no competing interests.

## Additional information



