## [Peer Review File · Communications Chemistry]

Reviewers' comments:

Reviewer #1 (Remarks to the Author):

This article by Flamme et al explores the use of phosphate variants as protecting groups in enzymatic synthesis of nucleic acids. In my opinion, this is a very interesting approach to addressing an important issue in nucleic acid chemistry, that is, what kind of protecting group can be used on the 3' OH of nucleoside triphosphates to control polymerization. The use of a 3' phosphate is insightful and could potentially address many of the issues that have hindered the use of other protecting groups. Although the results indicate phosphate is unlikely to work in this role, the process and associated results are highly valuable and will be of interest to the chemistry community, particularly the chemical biology/bioorganic chemistry community, and should influence these to look towards other chemical groups found in natural systems as protecting groups. The manuscript is nicely written, very interesting to read and provides an appropriate set of experiments and results. I recommend it be accepted for publication in *Communications Chemistry* in its present form. The only thing I would suggest is using the term "protecting group" in the title to better attract potential readers.

Reviewer #2 (Remarks to the Author):

Flamme et al explore an interesting idea, that of an alternative, removable 3'-nucleotide protecting group that could be used for both DNA and XNA synthesis based on a 3'-phosphate (3'P) group. Efficient 3' protection and deprotection would be an important advance, in particular for programmable enzymatic oligonucleotide synthesis. The authors describe synthesis of 3'P-nucleotides and various primer extension experiments testing different polymerases, substrate, buffer and template combinations and analyse the products by gel electrophoresis and LCMS.

There are a number of problems with this manuscript detailed below

- 1) The figures and analytics are not comprehensive and lack rigour (in particular the LCMS assignments, where it is very unclear what is what. The origin of many additional peaks is obscure and not discussed, many of which are of higher intensity than the peak they refer to). This makes it difficult to agree with their conclusions. Method sections are missing (e.g. for docking, LCMS analysis).
- 2) The authors find that phosphorylated nucleotides, which should result in a single incorporation and stop of synthesis, unexpectedly result in a vast mixture of products. No thorough attempts are made to understand what is happening in the different reactions. Control experiments with e.g. ddNTPs are missing, making interpretation of the gels challenging.
- 3) It also remains unclear how pure the synthesized 3'P-nucleotides were to start with or their stability under the buffer conditions used (no HPLC traces with standards, included).
- 4) The alternatively synthesised nucleotides (e.g. cyanoethyl-protected phosphate) don't add any novel findings. Since the authors suggest a phosphatase activity, 3'-protecting groups not based on this bond might have yielded more insights.
- 5) The polymerase phosphatase activity is not conclusively demonstrated or characterized here,

based on the lack of controls. Such an activity would anyway not be a completely novel or unexpected feature based on the previous findings that polymerases can exhibit 3' esterase activity (e.g. Canard 1995, PNAS).

To conclude, the manuscript does not present any real novel or impactful findings and lacks in rigour and execution. I therefore cannot recommend publication in its present form even with major revisions.

Reviewer #3 (Remarks to the Author):

Flamme et al. describe a preliminary investigation into the use of 3'-monophosphate as a protecting group for the controlled enzymatic synthesis of DNA and XNA. The authors synthesized several 3'-monophosphate analogs of thymidine nucleoside triphosphate, both as DNA and as LNA, and explored the reactivity of these substrates in template-directed polymerase-mediated primer-extension reactions. The authors unexpectedly discovered a novel phosphatase activity that appears to be associated with some of their DNA polymerases. This observation may offer new insight into the mechanism of DNA polymerases but weakens the suitability of 3'-monophosphate as a transient protecting group for the controlled enzymatic synthesis of DNA and XNA.

Major Comment

A weakness of this manuscript is the unanswered question of whether the observed phosphatase activity is truly the result of a novel DNA polymerase function, as the authors claim, or if the activity is due to the presence of a contaminating phosphatase enzyme. This is a critical problem that directly impacts the use of monophosphate as a protecting group for controlled DNA synthesis. If the activity is caused by a contaminant that could be removed by purifying the polymerase to a higher level, then the original strategy may still be viable. If, however, the activity does in fact lie with the polymerase, then the monophosphate approach is unlikely to succeed. This question needs to be answered before this manuscript can be accepted for publication.

Minor Comments:

1. The authors should consider improving the quality of their gels. The DNA primer needs to be purified so that it appears as a single band, and some gels should be re-formatted to show all of the bands in the gel.
2. The FAM dye should be clearly indicated in the figures and/or figure legends.
3. Line 292 mentions that a $\Delta m/z$ of 15 could correspond to a methylation event, but no hypothesis is given as to how this would be at all possible.
4. The double banding observed in Figure 5 and the 15 Da excess mass are intriguing and should be explained.
5. The authors should indicate that the terminal transferase reaction is untemplated, since this is the only reaction performed without a template.

Reviewers' comments:

Reviewer #1 (Remarks to the Author):

This article by Flamme et al explores the use of phosphate variants as protecting groups in enzymatic synthesis of nucleic acids. In my opinion, this is a very interesting approach to addressing an important issue in nucleic acid chemistry, that is, what kind of protecting group can be used on the 3' OH of nucleoside triphosphates to control polymerization. The use of a 3' phosphate is insightful and could potentially address many of the issues that have hindered the use of other protecting groups. Although the results indicate phosphate is unlikely to work in this role, the process and associated results are highly valuable and will be of interest to the chemistry community, particularly the chemical biology/bioorganic chemistry community, and should influence these to look towards other chemical groups found in natural systems as protecting groups. The manuscript is nicely written, very interesting to read and provides an appropriate set of experiments and results. I recommend it be accepted for publication in Communications Chemistry in its present form. The only thing I would suggest is using the term "protecting group" in the title to better attract potential readers.

Response: We thank this reviewer for the very positive assessment of our manuscript. We have changed the title of our manuscript to the following:

On the possibility of using 3'-phosphate as a transient protecting group for controlled enzymatic synthesis of DNA and XNA oligonucleotides

Reviewer #2 (Remarks to the Author):

Flamme et al explore an interesting idea, that of an alternative, removable 3'-nucleotide protecting group that could be used for both DNA and XNA synthesis based on a 3'-phosphate (3'P) group. Efficient 3' protection and deprotection would be an important advance, in particular for programmable enzymatic oligonucleotide synthesis. The authors describe synthesis of 3'P-nucleotides and various primer extension experiments testing different polymerases, substrate, buffer and template combinations and analyse the products by gel electrophoresis and LCMS.

There are a number of problems with this manuscript detailed below

Response: We thank the reviewer for assessing our manuscript and the constructive comments to improve the quality of our manuscript.

1) The figures and analytics are not comprehensive and lack rigour (in particular the LCMS assignments, where it is very unclear what is what. The origin of many additional peaks is obscure and not discussed, many of which are of higher intensity than the peak they refer to). This makes it

difficult to agree with their conclusions. Method sections are missing (e.g. for docking, LCMS analysis).

Response: We thank the reviewer for this important comment. We have now included a thorough description of the method used for docking experiments (in the manuscript) and for LCMS analysis (in the supporting information). We do agree that LCMS analysis of products stemming from PEX reactions is not an easy undertaking. For this reason, we have analyzed all peaks, regardless of their intensity, in order to provide as much information as possible. This allows a better understanding of the nature of the reaction products, especially for complex gels such as those shown on Figure 5 of the manuscript and follows literature MS and LCMS protocols (see e.g. Curr. Protoc. Nucleic Acid Chem. 47:7.16.1-7.16.11 and Sci Rep 7, 6674 (2017), respectively).

We have now included the following sentence in the manuscript and cited both references:

Intrigued by these results, we analyzed the products stemming from the PEX reactions conducted with the combination of Taq polymerase and 3'-phos-dTTP 5 as well as that with Terminator and 3'-phos-LNA-TTP 10 by LCMS (Table 1, Figures S23-S28, and Supporting Information) using established protocols for products stemming from PEX reactions with natural and modified nucleotides.

2) The authors find that phosphorylated nucleotides, which should result in a single incorporation and stop of synthesis, unexpectedly result in a vast mixture of products. No thorough attempts are made to understand what is happening in the different reactions. Control experiments with e.g. ddNTPs are missing, making interpretation of the gels challenging.

Response: We thank the reviewer for this comment. We have included control reactions in all gels where one or two natural nucleotides were missing to evaluate 1) whether misincorporation of natural nucleotides could occur instead of incorporation of the modified nucleotide and 2) provide size markers to compare with the products stemming from PEX reactions with modified dNTPs. All these controls are necessary and provide valuable information. The use of ddNTPs represents indeed another useful control that could confirm where the expected n+1 products run. However, synthesis of a ddLNA-TP is a very lengthy and challenging undertaking that is beyond the scope of this study.

3) It also remains unclear how pure the synthesized 3'-P-nucleotides were to start with or their stability under the buffer conditions used (no HPLC traces with standards, included).

Response: We thank the reviewer for this comment. All the modified nucleotides were purified by HPLC and thoroughly characterized by ¹H and ³¹P NMR. All the NMR spectra are in the supporting information and demonstrate the chemical integrity of all nucleotides. We have now included HPLC (anion exchange) traces of the modified nucleotides 5 and 12 in the supporting information (Figures S57 and S76) to further demonstrate the purity of these compounds. As stated in the manuscript and shown on Figure S17 of the original submission we have evaluated the stability of nucleotide 11 in TdT buffer at 37°C in the presence of the polymerase over time by LCMS. This analysis not only revealed that the cyanoethyl moiety was not removed during the process but also that the entire nucleotide 11 remained intact even after 12h. We have now performed additional stability experiments conducted in TdT buffer with or without TdT and with or without oligonucleotide at

37°C for three different time points (Figures S17-S19 of the revised ESI). All these experiments also demonstrate that nucleotide 11 is not degraded under these conditions. Collectively, these data clearly demonstrate the purity and stability of the nucleotides under primer extension reaction conditions.

4) The alternatively synthesised nucleotides (e.g. cyanoethyl-protected phosphate) don't add any novel findings. Since the authors suggest a phosphatase activity, 3'-protecting groups not based on this bond might have yielded more insights.

Response: We thank the reviewer for this comment. As stated in the manuscript, the presence of the two negative charges on the 3'-phosphate moiety is certainly responsible for the rather poor substrate acceptance by some polymerases under certain circumstances (e.g. as shown in Figures 2 and S5 of the initial submission). Therefore, we rationalized that the inclusion of a cyanoethyl protecting group would decrease the negative charge density and potentially increase the substrate acceptance. While we do agree with this reviewer that the addition of the cyanoethyl protecting group did not yield the expected effect, presumably due to increased steric bulk at the 3' position, it allowed us to address this important question and comfort us in the hypothesis that a promiscuous activity of polymerases might be responsible for the multiple incorporation patterns observed with templates containing multiple dA or randomized nucleotides. Similarly, nucleotides 15 and 16 modified with 3'-cyanoethylthiophosphate and 3'-thiophosphate protecting groups, respectively, were synthesized to verify whether the presence of a P=S center reduces the hydrolysis rate. While this did not yield the expected beneficial activity, these modified nucleotides also undergo a similar phosphatase activity which demonstrates its rather general nature. Taken together, while these compounds did not yield the expected outcome, they further validated the hypothesis of a novel type of catalytic activity mediated by polymerases. Other protecting groups have been evaluated (see e.g. references 39 and 40 of the original manuscript) and we have synthesized various nucleotides equipped with protecting groups that avoid ester linkages (i.e. phosphodiester and organic esters) to avoid esterase and phosphatase activities. Since we have made numerous analogues we feel that this large amount of additional data is beyond the scope of this manuscript and we will report the synthesis and properties of these compounds in due course.

5) The polymerase phosphatase activity is not conclusively demonstrated or characterized here, based on the lack of controls. Such an activity would anyway not be a completely novel or unexpected feature based on the previous findings that polymerases can exhibit 3'-esterase activity (e.g. Canard 1995, PNAS).

Response: We thank the reviewer for this comment. We have demonstrated that all modified nucleotides are pure and that polymerases do not sustain enzymatic synthesis when 3'-phosphorylated primers are used and therefore we believe that this phosphatase activity is truly catalyzed by polymerases. A more thorough, structural and mechanistic study will be required to shed light into the mode of action as well as the efficiency of this activity but we feel that this is beyond the scope of this present manuscript. We do agree with this reviewer that this activity is not totally unexpected since polymerases are known to exhibit 3'-esterase activity (see references 68-71

of the original manuscript) and we have changed a sentence in the discussion section to down tone this finding:

This observation is not totally unexpected since various polymerases including HIV-RT,...

Please also see our response to comments made by reviewer #3.

To conclude, the manuscript does not present any real novel or impactful findings and lacks in rigour and execution. I therefore cannot recommend publication in its present form even with major revisions.

Response: We thank the reviewer again for the evaluation of our manuscript and the constructive comments. We hope that the additional experiments, the inclusion of thorough descriptions of the methods used, and also the response to reviewer #3 will help convincing this reviewer of the quality of our experimental work. In addition, we firmly believe that the results disclosed in this article are of importance for the chemical biology community because we demonstrate that 1) phosphate displays all required properties to potentially act as a transient 3'-protecting group in controlled enzymatic synthesis (i.e. efficient blocking of synthesis with 3'-phosphorylated primers, ease of removal under mild conditions, small size, and biocompatibility); 2) 3'-phosphorylated DNA and XNA nucleotides can readily be obtained by our robust synthetic pathway; 3) despite the advantages highlighted in point 1), the negative charge and the newly disclosed phosphatase activity of polymerases preclude the use of phosphate as an efficient protecting group. In addition, this work represents the first effort towards controlled enzymatic synthesis of XNAs which opens up an important research avenue. Lastly, we believe that our compounds could be used to address other interesting questions in chemical biology such as why phosphorylation is used ubiquitously in biological systems but not to control enzymatic DNA synthesis and potentially in the fields of alarmones.

Reviewer #3 (Remarks to the Author):

Flamme et al. describe a preliminary investigation into the use of 3'-monophosphate as a protecting group for the controlled enzymatic synthesis of DNA and XNA. The authors synthesized several 3'-monophosphate analogs of thymidine nucleoside triphosphate, both as DNA and as LNA, and explored the reactivity of these substrates in template-directed polymerase-mediated primer-extension reactions. The authors unexpectedly discovered a novel phosphatase activity that appears to be associated with some of their DNA polymerases. This observation may offer new insight into the mechanism of DNA polymerases but weakens the suitability of 3'-monophosphate as a transient protecting group for the controlled enzymatic synthesis of DNA and XNA.

Response: We thank this reviewer for the positive assessment of our manuscript and the constructive comments.

Major Comment

A weakness of this manuscript is the unanswered question of whether the observed phosphatase activity is truly the result of a novel DNA polymerase function, as the authors claim, or if the activity is due to the presence of a contaminating phosphatase enzyme. This is a critical problem that directly impacts the use of monophosphate as a protecting group for controlled DNA synthesis. If the activity is caused by a contaminant that could be removed by purifying the polymerase to a higher level, then the original strategy may still be viable. If, however, the activity does in fact lie with the polymerase, then the monophosphate approach is unlikely to succeed. This question needs to be answered before this manuscript can be accepted for publication.

Response: We thank the reviewer for raising this very important point. Primer extension reactions with various polymerases as well as TdT-mediated tailing reactions with chemically synthesized, 3'-phosphorylated primers shown in Figures 1 and S1 clearly show that enzymatic synthesis is blocked. If contaminating phosphatases are present, they would be able to remove the 3'-phosphate moiety on the primers and restore DNA synthesis as shown in Figure S2. In addition, all polymerases that were used in this work were purchased from commercial suppliers. The supplier certifies that at least Vent (exo-), Kf (exo-), and Taq are devoid of any residual alkaline phosphatase activity, i.e. less than 0.0001 unit which is the limit of detection of the assay. However, in order to further address this question we have analyzed commercially available Kf (exo-) by MS (Figure S90 and Table S1). This analysis clearly reveals that this polymerase is devoid of any contaminant including phosphatases. Therefore, we strongly believe that multiple incorporation events are connected to a true phosphatase activity of the polymerases rather than contaminating phosphatases in the polymerase samples.

We have added a protocol on the MS analysis and details of the MS analysis of a control protein (Figure S89 and Table S1) and of the polymerase (Figure S90 and Table S1) in the supporting information and added the following section in the revised manuscript.

Certain DNA polymerases were recently shown to display an esterase activity (see Discussion) and hence a phosphatase activity is not totally unexpected. Even though commercially available DNA polymerases are certified by the supplier to display less than 0.0001 unit of alkaline phosphatase activity (New England Biolabs), we have performed an MS analysis on Kf (exo-) which confirmed the absence of any contaminants including phosphatases (Figure S90, Table S1, Supporting Information).

Minor Comments:

1. The authors should consider improving the quality of their gels. The DNA primer needs to be purified so that it appears as a single band, and some gels should be re-formatted to show all of the bands in the gel.

Response: We thank the reviewer for this comment. All primers and templates were purchased as HPLC-purified compounds from IDT. We have added a sentence in the Supporting Information to clarify this point:

All templates and primers were purchased HPLC-purified from Integrated DNA Technologies (IDT).

Most gels already show whether exonucleolytic degradation of primers occur. This is particularly important for Figures 1- 3 and S6 where PEX reactions were performed with 3'-phosphorylated primer P1, with both modified nucleotides 5 and 10 with the P1/T1 and P2/T2 systems, and in the

absence of dTTP. In these important examples, very little (<10%) degradation of the primer can be observed in all these figures.

2. The FAM dye should be clearly indicated in the figures and/or figure legends.

Response: We thank the reviewer for this comment. We have now clearly indicated in the text as well as in the caption of every figure of the manuscript and ESI that the primer used was located at the 5'-end.

3. Line 292 mentions that a $\Delta m/z$ of 15 could correspond to a methylation event, but no hypothesis is given as to how this would be at all possible.

Response: We thank this reviewer for this important comment. Indeed, methylation seems to be unlikely and we have removed this from the manuscript. Usually, such a $\Delta m/z$ of 15 results from misincorporation events such as incorporation of a dG nucleotide opposite dT (instead of dA) or dC incorporation instead of dT (see 10.1093/nar/26.11.2554). Yet only modified triphosphate was present. We have now changed this section to the following:

Such a $\Delta m/z$ is typically observed with misincorporation events (e.g. incorporation of dG instead of dA opposite templating dT or dC instead of dT opposite dA) under standard PEX reaction conditions. However, under our experimental conditions, only modified triphosphate 12 was present as substrate.

We have also inserted a citation to the following article:

Kirpekar, F. et al. DNA sequence analysis by MALDI mass spectrometry. *Nucleic Acids Res.* 26, 2554-2559, doi:10.1093/nar/26.11.2554 (1998).

For additional information, please also read our response to the next, connected comment by this reviewer.

4. The double banding observed in Figure 5 and the 15 Da excess mass are intriguing and should be explained.

Response: We thank the reviewer for this comment. Indeed, this is a very intriguing finding. So far, our best explanation is that the TdT phosphorylates the primer and the resulting products. Such specific phosphorylation mediated by the TdT has been reported for various modified nucleotides (*Nucleic Acids Res.* 2000, 28, 1276). While the pattern is near identical to this literature precedent,

the LCMS analysis of TdT-mediated products hints at a different modification event which yet remains elusive. We are currently working on experimental methods (including crystallization of the TdT with 3'-phosphorylated nucleotides) to try to shed light on this and we will report our findings in due time. We have now included the following sentences in the manuscript:

Similar double banding events such as that displayed in Figure 5 have already been described in the past for TdT primer extension reactions carried out in conjunction with sugar and 5'-phosphate modified nucleotides (10.1093/nar/28.5.1276). This gel pattern was ascribed to the capacity of TdT to phosphorylate (and phosphonylate) oligonucleotides. While the double-banding pattern appears similar, the LCMS analysis of products does not fit with such a phosphorylation event.

We have also included a citation to the following reference:

Arzumanov, A. A., Victorova, L. S., Jasko, M. V., Yesipov, D. S. & Krayevsky, A. A. Terminal deoxynucleotidyl transferase catalyzes the reaction of DNA phosphorylation. *Nucleic Acids Res.* 28, 1276-1281, doi:10.1093/nar/28.5.1276 (2000)

5. The authors should indicate that the terminal transferase reaction is untemplated, since this is the only reaction performed without a template.

Response: We thank the reviewer for this comment. We have now mentioned this on page 8 of the revised manuscript.

REVIEWERS' COMMENTS:

Reviewer #2 (Remarks to the Author):

The manuscript is now much improved, with a better characterisation of both substrates and products.

Reviewer #3 (Remarks to the Author):

I am satisfied that the authors have addressed all of my concerns.